# Predicting calving events in Antarctica using Machine Learning

Jacob Alexander Hay[*1], Hamzeh Issa[1], Daniele Fantin[1], David Parkes[2], Jan Wuite[3], Amber A Leeson[2], and Malcolm McMillan[2]

[1]S[&]T Norway
[2]Lancaster Environment Centre, Lancaster University, Lancaster, United Kingdom
[3]ENVEO IT GmbH, Innsbruck, Austria
hay@stcorp.no

## Abstract

Monitoring the calving dynamics of the Antarctic ice shelves is central to understanding a major driver for the changes to ocean levels on our planet. Several physical models have been proposed as calving laws, with varying predictive power. We propose an approach using Machine Learning (ML) to identify key variables and parameters that may be used in future models of the ice shelf calving dynamics. As part of an ongoing project, we have trained a U-Net on samples from a set of Gaussian Random Field (GRF)-represented Essential Climate Variables (ECVs). Ablation studies establish a few of the selected variables as having high correlation with calving events, with an F1 score above 0.9. Our first study site was the Larsen C Ice Shelf, on the northwest part of the Weddell Sea, where in 2017 there was a massive calving event. We have found strong correlations between the calving and the ice velocity leading up to this event, which may be further improved when accounting for basal melt rates in the area.

## 1 Introduction

The Antarctic continent is covered by a sheet of ice, known as the Antarctic Ice Sheet (AIS). Surrounding the outlets of the AIS are a series of ice shelves which are crucial indicators of the response of Antarctica to a changing climate, as well as being a major buttressing factor securing the grounded ice on the Antarctic [1, 2]. Collapsing ice shelves, where the whole shelf rapidly disappears, have a major effect on changing both the dynamics of the grounded ice as well as contributing to multitudes of downstream effects [3]. Predicting large scale events remains elusive to physics-based, process models, including the prediction of calving events. As part of the European Space Agency (ESA) project AI Forecasting for Ice Shelf calving (AI4IS), we have used a combined approach with a data cube of monthly GRF representations and machine learning to predict calving events, with the aim of providing a data-driven appRoach to forecasting ice shelf calving.

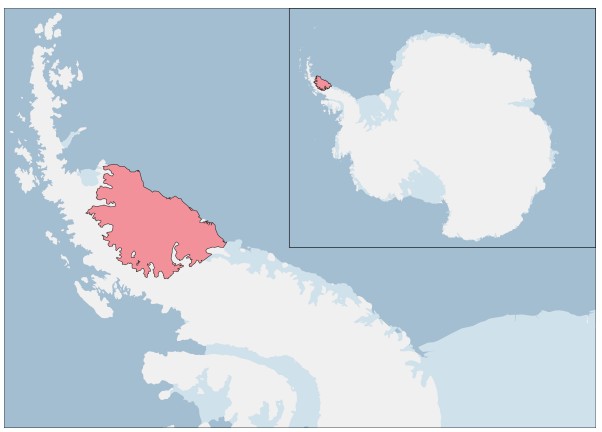

**Figure 1.** Location of the study area, Larsen C. Shown in red is the spatial data coverage over the ice-shelf.

Glaciers and ice shelves have been a field of study since the latter part of the 18th century: from the sliding dynamics of alpine glaciers under the force of gravity to looking at mechanical tensor descriptions of strain-fields, guided primarily by the search for physical laws determining glacier and ice sheet dynamics, and using numerical models to simulate the physical models since the 1950s and onwards [4]. Since the beginning of the 21st century, we have seen massive shedding of ice from the Antarctic ice shelves. Findings from the IMBIE assessment [5] report this shedding to be around 115 gigatonne per annum (Gt/a) in the period 2017-2020, with a peak discharge rate of 150 Gt/a in the prior 4-year period. Physical laws have been proposed to describe the process of calving [6] but, as noted by Wilner et al. [7], the majority of the proposed physical laws have remained unvalidated in the Antarctic.

In this paper we present the following contributions towards the study of calving dynamics in the Antarctic:

- We show that Deep Learning, and specifically U-Nets, can be used to predict calving events in the AIS, with up to a year of lead time before the event.

- We report on expert-in-the-loop validation, where model predictions were presented along-

---

[*]Corresponding Author.

Proceedings of the 7th Northern Lights Deep Learning Conference (NLDL), PMLR 307, 2026.

side eXplainable AI (XAI) heatmaps that showed model reliance on input variables, indicating that our models align with the current understanding of what are significant variables for determining future calving events.

- To the best of our knowledge, the model findings represent the first Deep Learning models to successfully forecast calving events off the AIS.

## 1.1 Related Work

In recent years, there has been efforts to better map the dynamics of the cryosphere by monitoring changes over time [8], and tracking the changing calving fronts [9, 10]. On the calving fronts there has been a circumpolar effort using a HED-UNet architecture by Baumhoer et al. [11], which has resulted in a fairly large, publicly available dataset. This was considered an option for automatic detection of label data, but ultimately was found to have too many misalignment artifacts to be useful. Recently there has also been increased efforts towards benchmarking calving front locations algorithms [12, 13], which we take as an indicator that the detection of calving front location will improve in the future.

Research in the field has been hinting towards the existence of universal calving laws [14], especially when allowing for stochasticity within the models. More extensive, full 3D physics models of calving have been made for Greenland [15], which show a reasonably close agreement between model and observed calving behavior, demonstrating primarily how our understanding of the physics has progressed. However, there still remains a way to go, as seen in systemic evaluations of calving models [16].

Meanwhile, in the Antarctic, efforts to model and predict the calving has been made using a data cube consisting of ECVs, and predicting calving with Random Forests [17]. This approach is often seen at early stages of spatiotemporal modelling, where one may wish to identify pixelwise properties that are indicative of some feature. The downside of such approaches is the loss of contextual information. Moncada [17] also indicated that the U-Net may be more favourable to the desired modelling.

U-Nets have been increasingly used for spatiotemporal, physical models. Here it is of particular note that standard U-Nets are comparable to physics foundation models [18]. Additionally, Tai et al. [19] have made progress towards formalising the mathematical explanation of the U-Net, showing that one can interpret the U-Net as a one-step operator splitting algorithm.

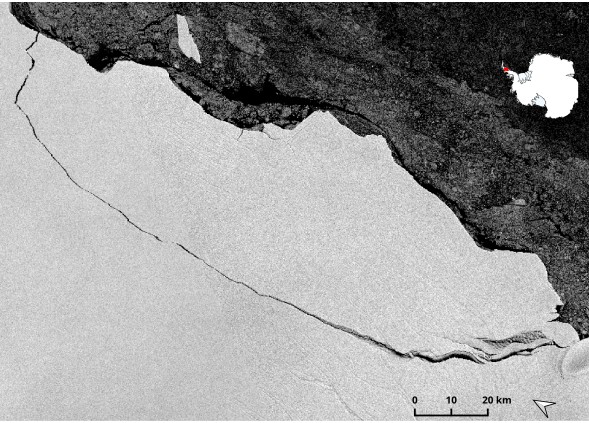

**Figure 2.** Sentinel-1 (SAR) image from 12/07/2017 of the A-68 iceberg calving event.

## 2 Method

This project has focused on the prediction of anomalous or extreme dynamics on the AIS, combining the use of a GRF-resampled data cube, and an ML model for approximating the transitions from slices of time to future states of known calving events on the Larsen C site (see Figure 1).

The Global Climate Observing System (GCOS) track collections of ECVs. Generally the ECVs constitute a list of factors that are essential for modelling and understanding the climate. Of these ECVs, GCOS has isolated a subset that pertains to the cryosphere in general, and to Antarctica specifically. In this project we have used an enriched set of inputs in addition to the ECVs that are commonly ascribed to ice shelves, but we will in this paper refer to all of the inputs as ECVs.

Our approach has been structured in these stages;

- **Data Collection** Data was collected and reprocessed into a GRF representation, and associated with known calving events as an (`input`, `output`) tuple.

- **U-Net training** A U-Net was trained to correlate the different slices of the GRF-data cube with calving events within a moving window of future events.

- **Significance analysis** Ablation studies has been used to identify the most significant contributions from the GRF-datacube, establishing the performance-metric correlation of each input.

- **XAI-based Saliency** XAI was used to generate an input-saliency map, highlighting the saliency of the inputs in a trained model.

- **Validation & Verification** Results of the ablation studies have then verified by domain experts to ascertain the scientific validity of the

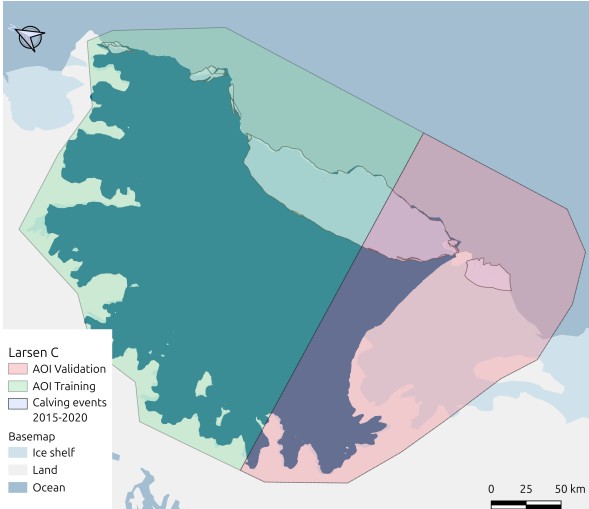

**Figure 3.** 2D surface map of our data. Showing data coverage (dark blue), training data (green) vs validation data (red), and calving events within 2015-2020 (bright areas). Calving events are here reproduced as vectorised data from Qi et al. [20], without spatial alignment and rasterisation.

predictions. Hold out validation data is used to track model performance (see Figure 3).

## 2.1 Data Cube

A multivariate datacube was constructed comprising various datasets relevant to ice sheet physics selected to cover the years 2014-2022 inclusive. Our first study-site was the Larsen C Ice Shelf, Figure 1, which had a large calving event in July 2017, where the approximately 1000 gigatonne (Gt) A-68 iceberg calved from the main ice shelf (Figure 2).

The datasets listed in Table 1 have been fitted and resampled to a GRF representation to standardise spatial resolution and fill data gaps. Noting that the coursest original resolution was the Wind Speed and direction (WS) at a resolution of $31 \times 31[km]$. Similarly both the Surface Mass Balance (SMB), and firn-related data were at a similarly course original resolution of $27 \times 27[km]$. These were therefore not expected to contain much information, but act as support for higher resolution data. The GRF was approximated at a set of irregular vertices on a mesh, before it was regridded as a contiguous surface of equal squares, i.e. as digital images. Samples taken from GRF have some process-based uncertainty, which was also produced as a gridded, spatially distributed output, and which will be quantified in the further analysis of the data cube, but which we for the purposes of our predictions have not taken into account.

It is important to note that the ECVs are assumed to have some level of correlation between them, which we implicitly use in our predictions.

**Table 1.** Table of datasets contained in the used version of Data Cube. All data is resampled from original resolution to 200 m resolution. The complete data cube will be published on a later date.

| Dataset (source) | Original Resolution |
|---|---|
| Ice Velocity (IV) [21] | $200 \times 200$ m |
| SMB [22] | $27 \times 27$ km |
| Firn thickness [23] | $27 \times 27$ km |
| Firn air content [23] | $27 \times 27$ km |
| Basal Melt (BM) [24] | $1000 \times 1000$ m |
| WS [25] | $31 \times 31$ km |

Such correlation would imply shared causality, which essentially means there is a pullback from the data-correlations to a common cause. We also make use of ablation studies to isolate individual ECVs, and identify whether these variables have strong correlations to calving events. Isolated variables in the ablations were also explored with XAI to highlight the saliency of the available inputs, estimate the validity of the model predictions.

In Table 1 we denote the included datasets available on the datacube at the time of the experiments. All datasets have been harmonised to a shared grid, with a shared resolution of 200-meter spatially, and a 1 month temporal resolution. The datacube was coupled with a corresponding 15 year calving dataset by Qi et al. [20]. This calving dataset was rasterised to the same resolution as our input data, and assigned values with an indicator function over the majority class in the corresponding vector data,

$$\mathbf{1}_{calving}(t) = \begin{cases} 1 \text{ - if calving at } t \\ 0 \text{ - if } \underline{not} \text{ calving at } t \end{cases} \quad (1)$$

To get prediction from the data, the time at which the labels are sampled were offset to the samples of the data cube. This was determined using a sliding window of data containing 3 months of input data, e.g. to validate prediction in July 2017 we provided data samples from April, May, and June of 2017 as our 3-month lead-time prediction[1]. Testing, and Figures shown here (Figures 6(a) and 6(b)) are from the month before the training and validation data (shown in Figure 4(a)), i.e. to predict an event in July we use test data from March. This strategy was selected to avoid the potential for hash-table overfitting error, where some models may learn an effective look-up table on low resolution data (see Table 1).

## 2.2 Sampling, preprocessing, and Augmentation

The data and labels were combined and gridded with overlapping grids to create the (`input`, `label`)

---

[1]ditto for 6 and 9 month lead-time prediction.

pairs which were randomly sampled by the models during training. All data was sampled at 200m per pixel, which translates to an XY-bounding box in the spatial domain of $1551 \times 1651$ pixels to cover the first study-site of Larsen C. Each training sample was targeted at $256 \times 256$ pixels. A Train/Test split was made based on two different sampling strategies (Figure 4), where a given "Lead Time"-parameter designated the temporal inclusion parameter for the data. Experiments were performed with either sampling strategy, showing little difference in performance. The numbers in Table 3 are from the "lead time"-strategy (Figure 4(a)).

All the data were split into `Training` or `Validation` by intersection of a vectorised Area Of Interest (AOI) (see Figure 3), where the major calving event of A-68 is roughly intersected to reserve $\frac{1}{3}$ of its total area for validation. The splitting of the ice-shelf remains the same throughout the time-dimension, to ensure no potential learning of the precursive configurations, or memorisation of the outlines carries over from training to validation data. The remaining ice-shelf is similarly split using vectorised areas, and masking areas that do not intersect with the target AOI.

Given the relatively low amount of valid calving events, we employed a stack of random augmentations. For this the sample sizes of the training samples was scaled up by a factor of $\approx \sqrt{2}$ to $364 \times 364$, in order to account for full rotational freedom without losing information when clipping to the target shape. These samples were then rotated by a random angle up to $\pm 45°$. After rotation samples were randomly flipped along both horizontal and vertical axes, with a probability of 0.5, before being cropped to the target input shape of $256 \times 256$. This was done to improve generalisation, and to increase the amount of training samples.

## 2.3 Model and Parameters

We have used an attention U-Net [26, 27] architecture for our model. U-Nets generally produce readable maps as outputs, and their spatiotemporal properties are well supported, as is evident by them serving as a baseline for comparison in physics foundation models [18]. Derivatives of the U-Net also has a long standing, empirical track-record classifying remote sensing data, and consist of an encoder, a decoder, and a bottleneck, with skip-connections to act as control-parameters for the reconstructed image of delineated features in the output map. Furthermore, as shown by Tai et al. [19], U-Nets are effectively solving a control problem. In our case, this means they approximate the transition maps inherent to our causal model, decoded as a probability-distribution over calving events. Our U-Net was trained using an AdamW optimiser [28], and following recent rec-

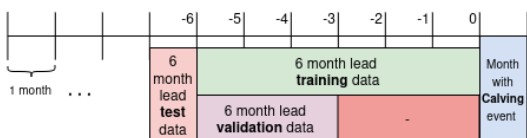

**(a)** Lead time based sampling strategy. Example shown for 6 month lead time, using a 3 month window for validation. Inclusion of training data for the full period was intended to provide some lifting. Limiting the validation data to the first 3 months of the available lead time was intended to ensure model selection trends towards further reaching prediction within the lead time window.

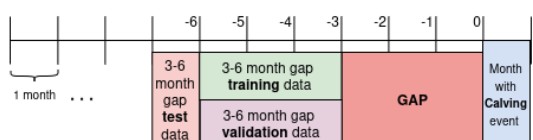

**(b)** Gap time based sampling strategy. Similar approach as lead time, using only the first 3 month window for the targeted prediction lead times.

**Figure 4.** Time-wise sampling strategy for data, with 6 month lead example. Colours correspond to the spatial AOI split in Figure 3. On the right showing the month containing a calving event, and on the left showing the temporal segregation of the test data. Resulting images from models run on test data are collected in Appendix A

ommendations from the study on hyperparamters by Orvieto and Gower [29] we set $\beta_1 = \beta_2 = 0.9$.

We have used Focal Loss [30] as our target for classification, with $\gamma = 3$, to account for some of the class imbalance, and our target metric was the macro-averaged F1-score, which should avoid biasing the metric towards the `no-calving` label. We used weighting in favour of `calving` in both the random grid-sampler and the loss function, with a class-weight of $2 : 1$. For ease of reference see Table 2.

## 2.4 Ablation & XAI

We have performed a set of ablations over the available dataset, focused on identifying the strength of the correlations between the available input data from the data cube and the labels from Qi et al. [20]. To do this we sliced the data cube into its constitute parts, and trained U-Nets for each variable in turn, under similar conditions. The ablations were performed across a selection of lead-times, divided into sections of 3-month periods, and predictions evaluated on the final time-slice that was not included in the training data. This was done to ensure the data was representative of the region trained on, but not included in the training data. Early results indicated that IV had the strongest correlation with calving, followed by BM.

Therefore, we performed an additional experiment using a combination of the two most predictive subsets of the data: IV and BM (Figure 6(a)). It should

**Table 2.** Attention U-Net[27] trained with AdamW, model parameter summary:

| Model | |
|---|---|
| Model | Attention U-Net |
| Activation | ReLU |
| Encoder Blocks | 5 |
| Base channels | 16 |
| Output channels | 2 |
| **Optimizer** | |
| Algorithm | AdamW [28] |
| Learning Rate | 0.001 |
| $\beta_1$ | 0.9 |
| $\beta_2$ | 0.9 |
| Weight Decay | 0.1 |
| **Loss** | |
| Loss Function | Focal Loss [30] |
| $\gamma$ | 3 |
| classes | `(no-calving,` `calving)` |
| class weights | `[1, 2]` |
| reduction | `mean` |
| **Data Augmentation** | |
| Input-shape | $364 \times 364$ |
| Rotation | $\theta \in [0, \pm 45°)$ |
| P(Horiz.flip) | 0.5 |
| P(Vert.flip) | 0.5 |
| Center Clip | $256 \times 256$ |

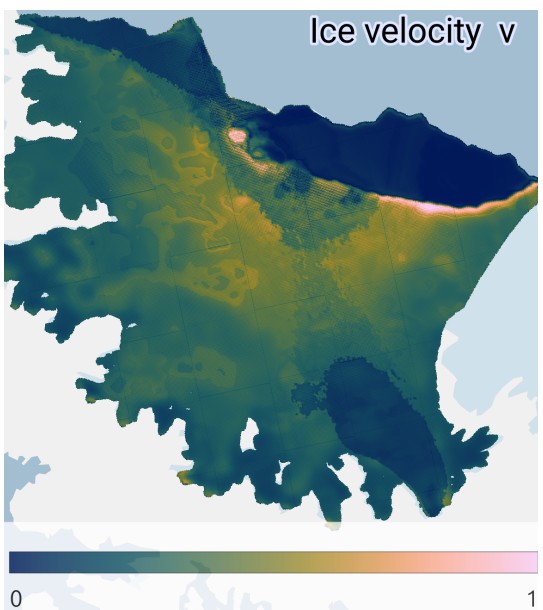

**Figure 5.** Cropped detail image from a XAI saliency map generated for a model trained with IV and BM as inputs. Note also the delineation close to the boundary of the calving area. Full version of image is included in Appendix B

here be noted that the basal melt data is partially dependent on the ice velocity data. Combination of ablation and XAI was used to identify key input variables. To establish the significance of the different layers in the data cube, we divided the dataset. For the ablations we started with the individual ECVs, and measured the performance. Consecutive experiments looked at the combination of variables within $\mathcal{P}(ECV)$, of which we here report only on the best performing combination and the individual subsets.

After we identified the contribution of each ECV to the final performance, we have used guided backprop, and a variation of GradCAM [31] to highlight the area of the input that is most salient for the predicted output (Figure 5). Verifying the areas the model finds salient acts as a constraint on the applicable mathematics that goes into considering whether the model seems to cover existing theory, or if it may be completely confused.

## 2.5 Validation

For each ablation experiment, the model predictions, metrics and data were prepared for the entire ice shelf, focusing primarily on the validation area of Larsen C. The quality of the predictions was then validated by domain experts from Centre of Excel-

lence in Environmental Data Science (CEEDS) at Lancaster University (LU), who made a qualitiative verification of whether the causal relationship fits with our current understanding of glaciology.

This form of validation was considered a more appropriate approach to the standard train-test split of data primarily due to the limited amount of calving events observed from Larsen C during the period of available data. Secondly, since a stated goal of the project was to identify clear links between the data and the predictions of our ML approach, the validation of these causal predictions require some form of human-expert-in-the-loop. This provides necessary feedback to ensure that our models are more likely to rely on real physical relationships rather than data correlations that have no connection to the actual dynamics of the system. Finally, we have provided a GradCAM [31]-based XAI saliency map over the multivariate data cube (example in Figure 5). These were intended to inform domain experts about the relative weight the ML model places on the spatially distributed information of each channel. The validation of these also require familiarity with the inputs given to the model.

## 3 Results

### 3.1 Ablation

Table 3 lists the F1-scores of the ablation experiments with different datasets along with their respective lead times. We found that IV both with

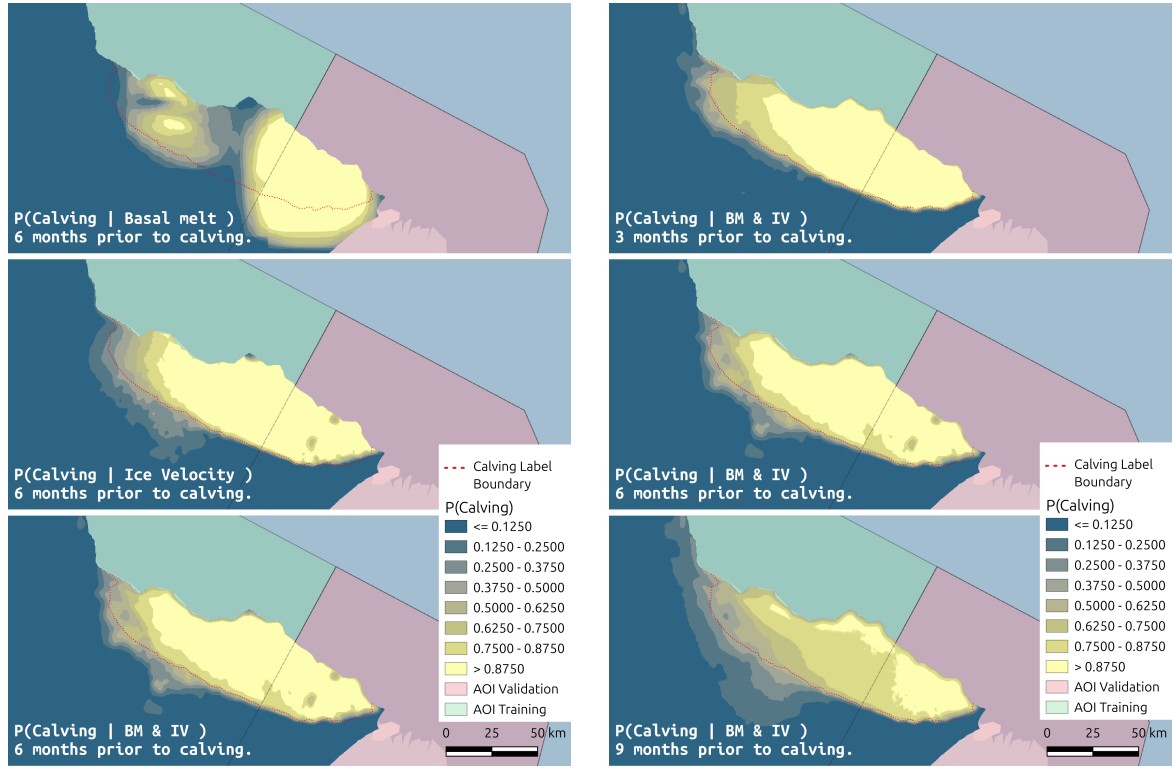

**(a)** Predictions with varied subsets of data

**(b)** Predictions with varied lead-time.

**Figure 6.** Differences in predictions. Figure 6(a) shows the predictions from models trained on subsets of the available data; basal melt rate, ice velocity, and a combination of basal melt and ice velocity, respectively. Figure 6(b) shows the effect of temporality, with the models given a 3, 6, or 9 month lead time of data.

and without the BM have strong correlations to the target labels (shown in Figure 6(a) and Figure 6(b)). No predictive power was found with the firn air content. Firn thickness showed some promise as a potential supporting variable, but had unreliable predictions on the test set. Similarly we identified the SMB to be a potential candidate for supporting variables, but ultimately not as useful as the BM for supporting models that make use of IV.

WS seemed to correlate more than expected with the calving, both on training and validation data, but the higher F1-scores reported only emerged towards the end of the trainings. When predicting on the test-data the correlations seemed to fully disappeared. For the Test-maps presented to expert validation, the model predictions did not contain discernable or significant information. This may therefore have been case of overfitting, where the model associates the values of the wind speed directly to the label rather than generalising; a hypothesis further supported when considering the original resolution of the wind speed data (Table 1).

## 3.2 Validation

Predictions from ablated models have been presented to domain experts from the polar science group at LU associated with the CEEDS and Centre for Polar Observation and Modelling (CPOM). Model performance is assessed with reference to experience with observed calving events and understanding of the physics behind the calving process [7, 32].

The results on the variables with most predictive power were found to be consistent with an understanding of the physics of ice shelf calving. Ice velocity, the most significant predictor, has a direct relationship with calving; for floating ice closer to the calving front and further from the grounding line, high variability in velocity over short distances suggests it is associated with rifting, cracking, or other forms of disturbance to otherwise smooth ice flow which are significant in the process of icebergs detaching from the shelf.

Basal melt as the second most significant predictor has a major impact on ice shelf thickness, with significant thinning of the shelf making calving easier. It is not surprising that the other variables used do not appear to have major predictive power. They have a less direct impact on the calving process, so despite likely having some information to impart about ice shelf stability it is not expected that any of these datasets individually allows for confident prediction of imminent calving. Surface mass balance is not the dominant factor in mass changes for Antarctic ice shelves, with basal melt and calving much larger sources of mass loss. Changes in firn

**Table 3.** Highest validation F1-score mean vs lead time. Highest F1-scores per lead time highlighted in **bold**. Values reported are the Mean (M) of highest F1-scores and their respective Standard Deviation (SD).

| Subset | Lead time | |
|---|---|---|
| | **3 months** | **6 months** |
| IV | **0.951** ± 0.015 | 0.936 ± 0.0043 |
| BM | 0.865 ± 0.0097 | 0.822 ± 0.024 |
| IV + BM | 0.941 ± 0.0036 | **0.937** ± **0.0025** |
| SMB | 0.605 ± 0.11 | 0.622 ± 0.18 |
| WS | 0.786 ± 0.02 | 0.739 ± 0.02 |
| firn thickness | 0.550 ± 0.1 | 0.596 ± 0.14 |
| firn air content | 0.491 ± 0.005 | 0.486 ± 0.009 |

| Subset | Lead time | |
|---|---|---|
| | **9 months** | **12 months** |
| IV | **0.927** ± **0.0085** | **0.945** ± **0.0073** |
| BM | 0.785 ± 0.04 | 0.794 ± 0.0086 |
| IV + BM | 0.926 ± 0.0019 | 0.928 ± 0.0032 |
| SMB | 0.488 ± 0.0065 | 0.506 ± 0.011 |
| WS | 0.749 ± 0.009 | 0.778 ± 0.01 |
| firn thickness | 0.623 ± 0.14 | 0.630 ± 0.01 |
| firn air content | 0.487 ± 0.006 | 0.487 ± 0.0005 |

thickness and firn air content are also indications of surface processes and are not expected to have a large impact on overall flow. Wind speed may have some direct and indirect impact on ice shelf calving, but the coarse resolution of the input dataset and the fact that wind speed variations aren't likely to be localised to areas prone to calving explains the relative lack of predictive utility.

The spatial predictions of calving location (Figure 6(b)) matches well with the geometry of the A-68 calving event. The closer contours in the southern part of the domain (right of Figure 6(b)), suggesting higher confidence in the location, are consistent with the way the calving event played out, with the separation beginning in this area and propagating north (left of Figure 6(b)).

## 4 Conclusion

In this work we have shown that a U-Net can be used for predicting future events, acting as an approximation of the temporal transition map. While these predictions are now at a stage that shows clear correlations, it still remains to see how well they will perform on more rapid ice shelves. It should also be noted that the correlations are currently statistical correlations, and not strongly founded in physical theory. A further review of proposed physical models for the calving dynamics of the AIS can be made on the basis of these correlational models, where clear correlations such as the ice velocity are more closely considered.

We have found that our models can train to a high degree of correlation for long-reaching causal chains, as measured by the F1 scores for lead-time data. The question of generalisability remains open, with the model currently only being validated for a relatively stable ice sheet. Further modifications to the experiment design may provide more insight, and we are considering a change from the lead-time approach into a windowed gap-based approach.

Confirmation that the model performance remains consistent with current understandings of the physics of ice shelf calving is promising. This indicates the models used here may be good approximations to the current physical models for ice shelf dynamics. Future work will extend this work to encompass larger areas of the Antarctic, exploring possibilities of using transfer learning between different ice shelves, as well as more dynamic and faster flowing shelves.

## Acknowledgments

This work was funded by the ESA as part of the AI4Science ITT (ESA Contract No 4000143299/23/I-DT).

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

# A    Appendix: Figures

## A.1    3 month predictions

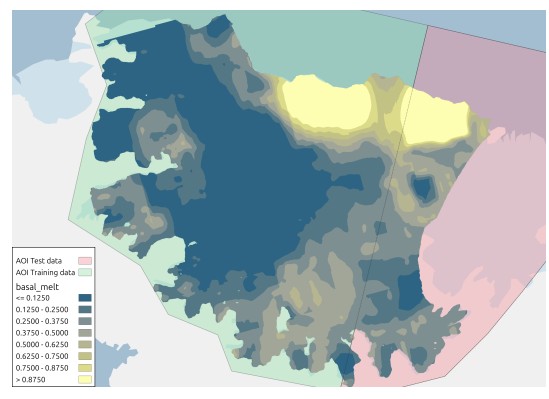

**(a)** 3 month lead, BM

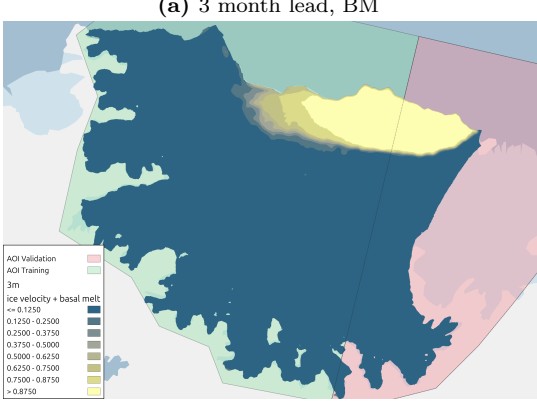

**(b)** 3 month lead, IV and BM

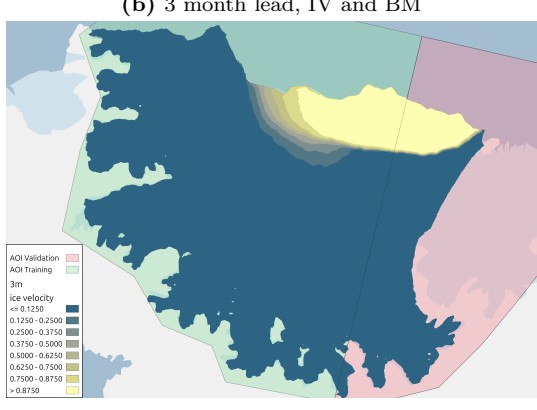

**(c)** 3 month lead, IV

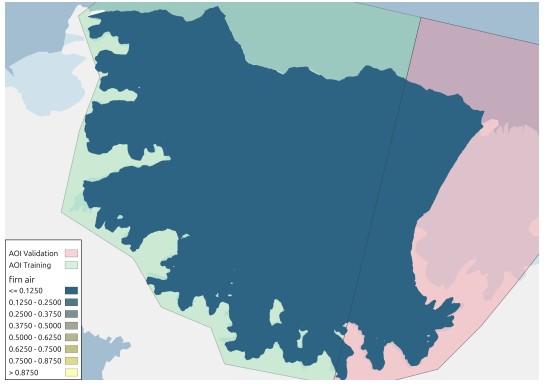

**(a)** 3 month lead, firn air content

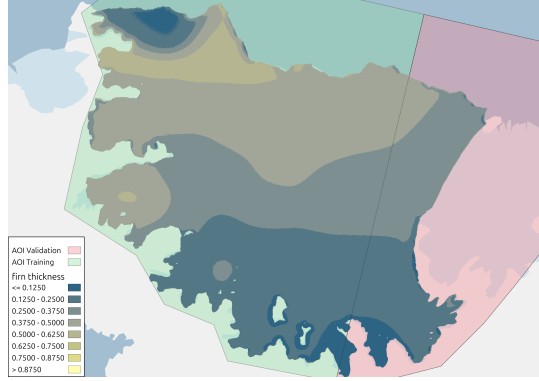

**(b)** 3 month lead, firn thickness

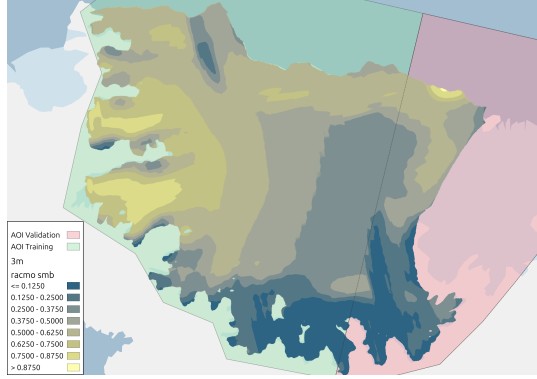

**(c)** 3 month lead, SMB

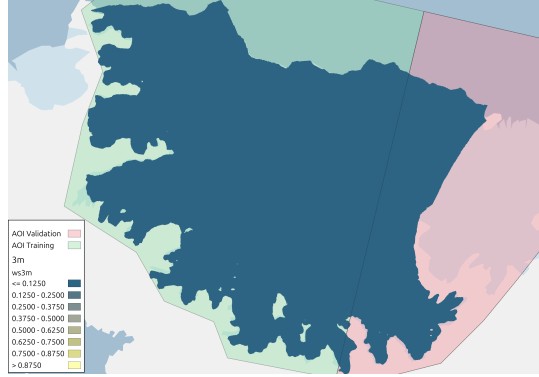

**(d)** 3 month lead, WS

## A.2 6 month predictions

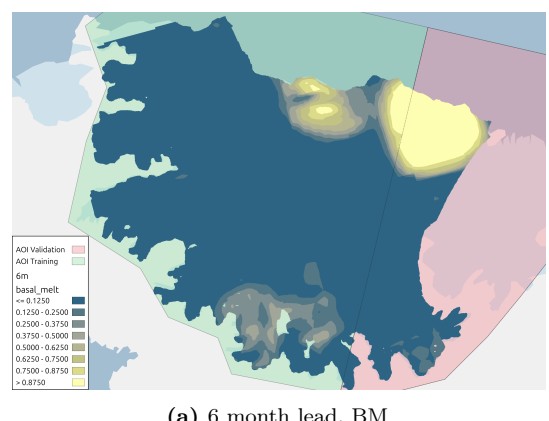

**(a)** 6 month lead, BM

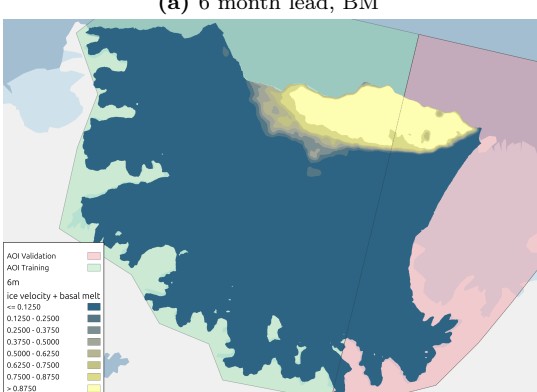

**(b)** 6 month lead, IV and BM

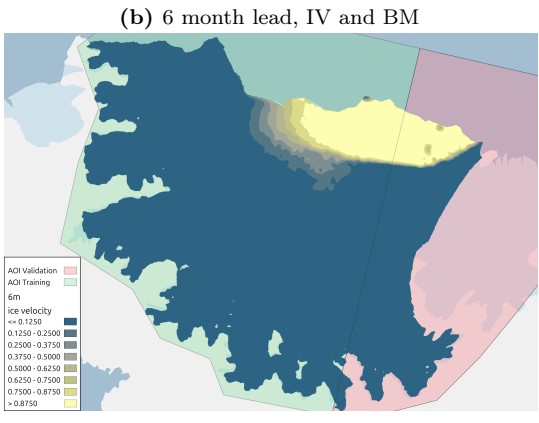

**(c)** 6 month lead, IV

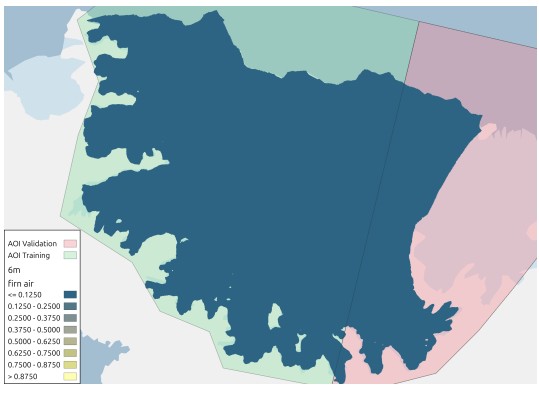

**(a)** 6 month lead, firn air content

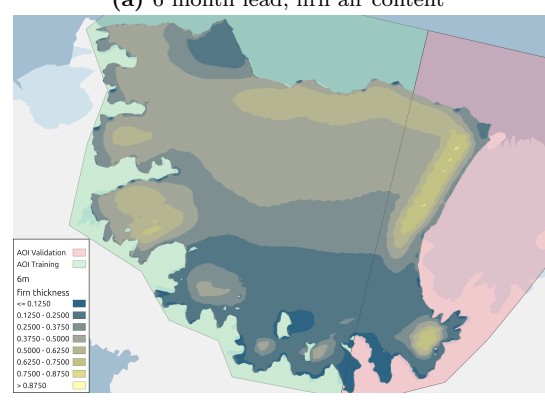

**(b)** 6 month lead, firn thickness

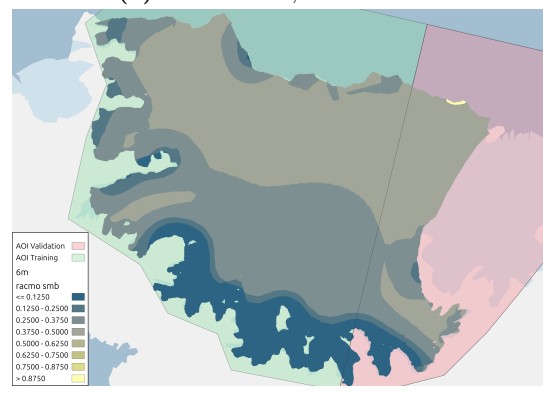

**(c)** 6 month lead, SMB

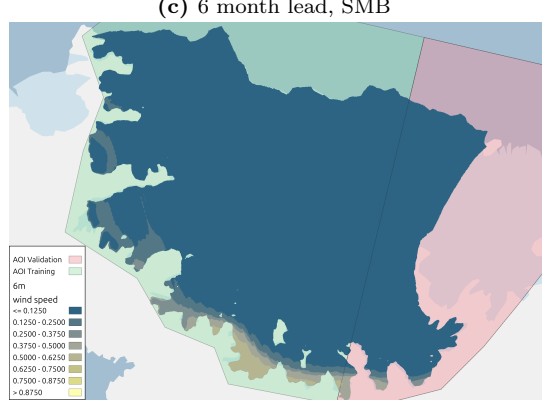

**(d)** 6 month lead, WS

## A.3 9 month predictions

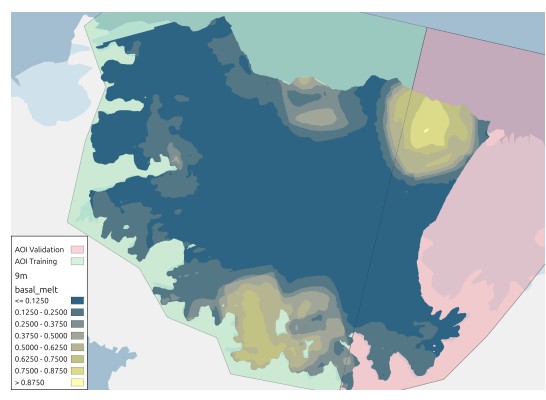

**(a)** 9 month lead, BM

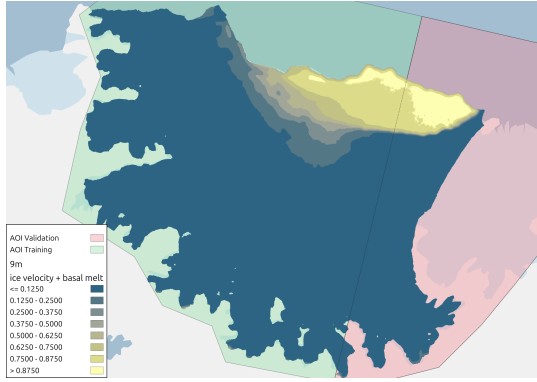

**(b)** 9 month lead, IV and BM

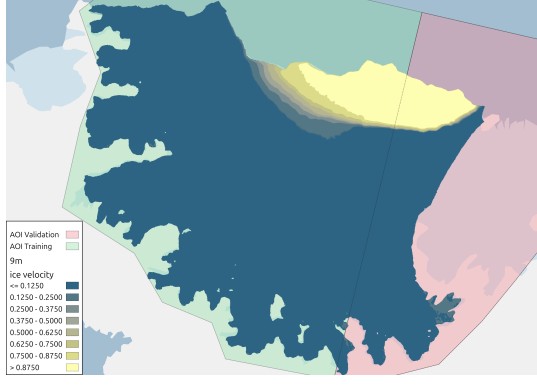

**(c)** 9 month lead, IV

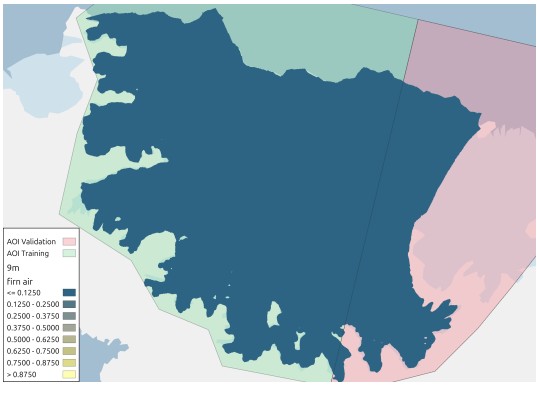

**(a)** 9 month lead, firn air content

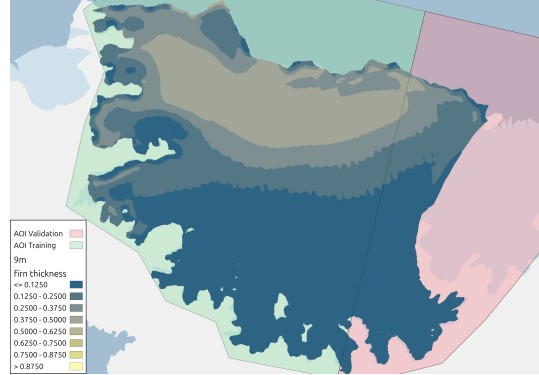

**(b)** 9 month lead, firn thickness

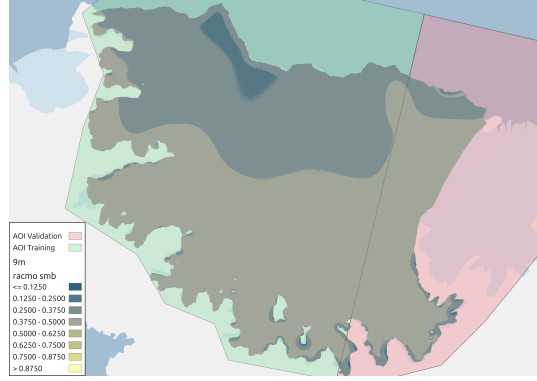

**(c)** 9 month lead, SMB

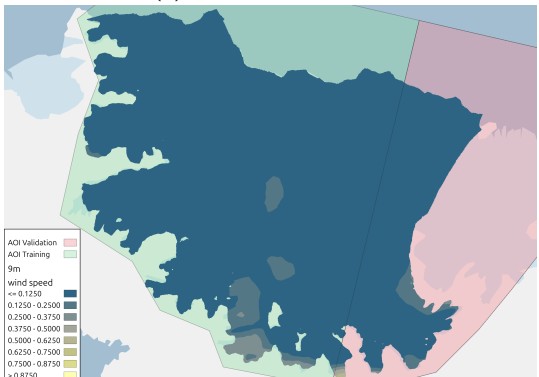

**(d)** 9 month lead, WS

# B  XAI Saliency Map

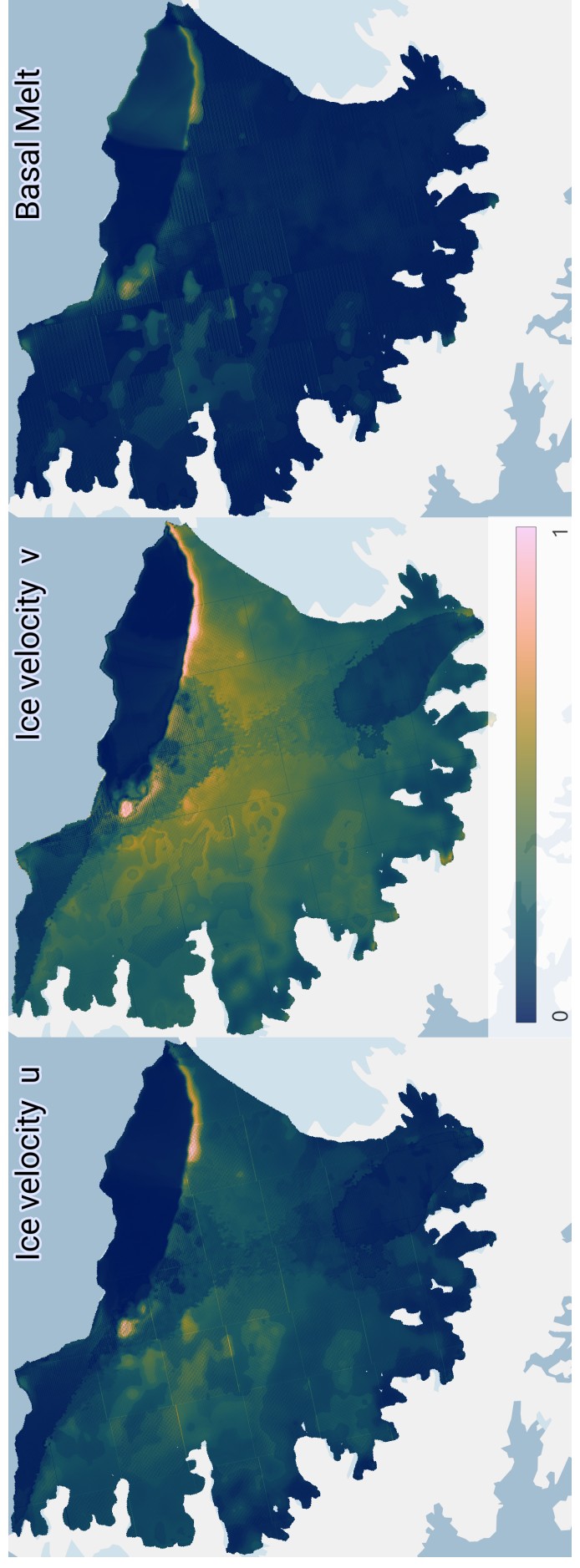

**Figure B.1.** Merged full Ice Shelf view of XAI saliency map generated on a model trained with IV and BM as inputs, showing a relatively higher saliency for ice-velocity. Note here the tendency to place higher saliency close to the boundary of the calving event.

