# OpenReview forum: "Predicting Calving Events in Antarctica using Machine Learning"
_NLDL.org/2026/Conference — NLDL 2026 Poster_

### Official Review · Reviewer_vmBi · 2025-10-05
**Interesting but somewhat limited study on deep learning for predicting calving events in Antarctica**

**Rating:** 4
**Confidence:** 4

**Summary:**

This work is focused on predicting calving events in Antarctica based on climate variables using a deep learning approach. A multivariate data cube is collected and input-output pairs are constructed. The performance of the deep learning models is evaluated across different time horizons and for different input variables. Results show that the proposed deep learning approach can provide strong performance, thus paving the paving for a potential new use case of deep learning.

**Strengths:**

1.  Using deep learning for predicting calving events appears to be a new application area, with potential for impact in an important domain for climate science.

2. The data gathering, construction, and setup undoubtably requires a great deal of work and can serve as an important starting point for future works along the same lines.

3. The paper is clearly motivated and well-written.

**Weaknesses:**

1. The experimental evaluation is somewhat limited. In particular, the introduction mentions that "Predicting large scale events remains elusive to physics-based, process models,", but it would have been highly beneficial if such a physics-based process model was implemented and applied to give a baseline. Without it, it is difficult to say how effective the deep learning models is.

2. Similar to the point above, it would be useful with a simple machine learning-based baseline to demonstrate if a highly computationally demanding deep learning approach is necessary for this particular task. For instance creating summary statistics from the data cube and training a simple linear algorithm of some sort.

3. The quantitative results are presented without any indication of variation across training runs, which limits the experimental evaluation [1]. Providing e.g. the standard deviation across multiple training would give the reader an impression how stable the performance of the model is for this particular task.

[1] D. Trosten, Questionable Practices in Methodological Deep Learning Research, NLDL 2023.

**Justification:**

The combination of deep learning and climate variables for climate-related tasks is not something new (see e.g. [1]), but I believe this is the first application for this particular task. That makes it interesting and provides potential for impact. Furthermore, the experimental evaluation is limited, particularly in terms of missing baselines. If some kind of simple baseline could be provided it would greatly strengthen the paper. Nevertheless, I think the combination of a novel application and a clearly written paper still makes this an interesting work that can spark discussions at the conference, and I am therefore leaning towards acceptance.

[1] Andersson et al, Seasonal Arctic sea ice forecasting with probabilistic deep learning, Nature Communications, 2021

---

> ### Author Rebuttal · Authors · 2025-10-20
>
> Thank you for your review of our work, and for the referenced to Andresson et al.
> We will include this in our revision, as part of a Related Work-section.
> Regarding the weaknesses you point to:
> - We will try to reach out to potential partners who may have implementations of physical models suited for these predictions to compare with. However it is unclear now if there will be anything available on short notice for an eventual revision to this paper.
> - Point taken, we could better justify the usage of deep learning by comparing with simpler ML models. We can try to run a simpler model with RF or Boost as comparision baseline.
> - This is valid. We will run a few more trainings from different seeds and include reporting of the variation in the revision of the paper.
>
> Ps. The reference to D.Trostens paper is much appreciated. We will make further attempts to correct for the issues they identified in work performed, and work these questions raised into our processes going forward.

---

### Official Review · Reviewer_t1tR · 2025-10-07
**Good, but lacks summary of related work**

**Rating:** 2
**Confidence:** 4
**Final Rating:** 4
**Final Confidence:** 4

**Summary:**

The authors present their results from training an AI model to predict ice calving events in the antarctic. They describe how they prepared the training and input data, which model they've used, and how it performs in practice. The latter is evaluated by applying the model to a study site and through a qualitative validation by domain experts.

**Strengths:**

* The paper is very well written and easy to understand
* The topic is of high interest and definitely worth investigating
* The work fits very well into the conference topics
* The method is sound and the evaluation is reasonable

**Weaknesses:**

* I have only one major issue with this paper: It lacks a section on related work. Without this, it is very hard for the reader to judge if the paper provides a reasonable scientific contribution or not. I know the authors say they've done a comprehensive literature study and were unable to find a comparable work, but in my opinion, this is not enough. To some readers (not necessarily me, since I find the rest of the paper very compelling), this might sound like the authors were lazy or just did a bad job at reviewing the literature. "Related work" does not necessarily mean you have to find other works that did the exact same thing as you, it's more about finding work that does similar things (or at least in some aspects) but in a different way or applied to a different application domain, for example. A related work section is also a good way to show the reader where the research gap in literature is and how you try to fill this gap with your work (essentially, in which way your work excels). In the case of this paper, I would imagine that other people have tried predicting calving events in the past, but all previous methods were too tedious or too error-prone. Or, maybe somebody else has applied a UNet to temporal data, not to predict calving events, but for something else, and was also successful, which could be interesting for other readers. There is still enough room to include a related works section. There is white space on pages 5 and 8. Also, see next two bullet points.
* Section 5 on future work seems kind of redundant to me. After I've read Section 4, it did not add anything new in my opinion. It can safely be removed.
* Reference [7] has a very long list of authors. After the 7th name, the remaining names can just be replaced by an ellipsis "..." or "et al.".
* In scientific work, when referring to figures and tables, the first letter of “Figure” and “Table” is usually capitalized.
* Also, in scientific work, contractions are usually avoided (see "It's" on page 2, for example). Please revise.

**Final Justification:**

The authors addressed my concerns in their rebuttal. If they revise the paper accordingly, it can be accepted in my opinion. Thank you very much for the effort!

**Justification:**

The paper would be an accept if there was a comparison with existing works. Without this, it's very hard to judge, if the contribution warrants publication.

---

> ### Author Rebuttal · Authors · 2025-10-20
>
> Thank you for your feedback.
> We will include a section "Related Work", following your recommendation. There is not a whole lot, but there are works performed on e.g. calving front detection, and another reviewer brought up an earlier attempt at using a UNet to perform similar work. Your point on available white space is well taken, so we will make a fuller version that properly adresses these concerns.
> We may remove the section on Future work from our revision, allowing the Conclusion to conclude.
> Yes, this reference was very long, it was unclear from the author specification whether we could alter the formatting of the references, so it was left as is. We can replace this with "et al" after the 7th.
> We will fix the capitalisation of references in the revision.
> Contractions to be revised, and we will try to adhere better to conventions of scientific publications in the future.

---

### Official Review · Reviewer_CjxL · 2025-10-09
**Important application and interesting findings but lacking details and not reproducible**

**Rating:** 2
**Confidence:** 4

**Summary:**

The paper presents a framework to predict ice shelf calving in Antarctica, specifically focusing on the Larsen-C ice shelf which produced the large iceberg A-68 through a calving event in 2017. The paper trains a variety of U-Net models on Gaussian Random Field representations of essential climate variables known or hypothesised to have an impact on ice sheet calving events, with the aim of developing a predictive model of calving events at lead times of 3, 6, and 9 months. The paper finds that ice velocity and basal melt rate are the two most informative predictors in an ablation study and achieve high F1 scores on a binary classification task of whether or not the ice shelf under consideration will calve at a given lead time. The resulting maps of calving probability are validated by a panel of domain experts who confirm that the model is picking up on physically realistic signals in the data. The paper grounds the approach in a stochastic dynamical systems frame of view with connections to causality.

**Strengths:**

The paper's strengths, and how these tie into expected criteria, are summarised below:
1. Important application - the phenomenon of ice shelf calving is not well predicted by any major models at this time, yet plays a large role in the stability of the Antarctic and Greenland Ice Sheets and plays a large role in downstream impacts including sea level rise. The paper contributes towards the scientific community's knowledge of calving event predictability and the variables which are useful in making such predictions. In this aspect, criteria "11. Significance" is upheld.
2. Application-specific data fusion and modelling considerations - the paper makes sensible choices when selecting predictive climate variables, choosing reasonable data augmentation techniques, selecting the use of Focal Loss when working on a class-imbalanced classification problem, and breaking the region into overlapping windows. In this aspect, "3. Application Papers" is met, with the non-trivial challenges and the proposed solutions to these challenges arising when working with environmental data being well explained.
3. Working with domain experts - the fact the results of the paper are discussed and receive a "seal of approval" from domain experts in ice sheet dynamics is encouraging, tying into criteria "1. Correctness", and "12. Attribution"

**Weaknesses:**

The paper's weaknesses, and questions to the authors to be addressed during rebuttal, are summarised below:

1. Data preprocessing steps are incredibly unclear, significantly hampering the paper's reproducibility (criteria 4)
1.a. It is mentioned that the data cubes are represented as a set of Gaussian Random Fields, but no further detail is provided regarding this representation.
1.b. It is not clear whether any normalisation scheme is applied to the data before being passed into the model; with physical variables taking large values being considered, surely some sort of normalisation scheme must be applied to achieve good results?
1.c. What size tiles, or data samples, are considered? How many? This is important to assess the context window size considered by the U-Net, and I would imagine the results could change significantly depending on how large/small tiles are considered.
1.d. What train/test split is considered? Is the data considered only that leading up to Larsen-C? If so, how can the results be considered generalisable? Or are they only meant to be a case study? It is mentioned briefly that "This form of validation was considered a more appropriate approach to the standard train-test split of data primarily due to the limited amount of calving events observed from Larsen C during the period of available data," but this does not provide sufficient information for one to reproduce the training/validation strategy.
1.e. No code or data cube is provided, though it is mentioned that the "complete data cube will be published on a later date."

2. Connections of experiments to theoretical framework presented in the introduction are weak, confusing the presentation and calling into question certain elements of experimental rigor (criteria 5, 8)
2.a. A significant point is made to introduce the language of stochastic dynamical systems in the introduction, but this language is never used throughout the rest of the paper, obscuring (in my opinion) the main narrative and results of the paper. Either the mathematics should be referenced throughout, or the dynamical systems introduction could be removed (I would suggest the latter in favor of more details on experimental setup, as above in point 1). It is claimed that the ML models trained are modelling a observed dynamical map through time but it is not clear how this is helpful, exactly.
2.b. The language of causality is thrown around in the paper without proper mathematical treatment. To the authors' credit, three criteria for causality are presented in the introduction and connected to the dynamical systems point of view, but the modelling approach does not include any specific techniques from causal machine learning and seems to be using the language of causality to aggrandize the results of a predictive model. The paper is not doing causal inference, and while there are real physical connections between the predictor variables and the observed calving events, I'm not convinced the language of causality is helpful in the presentation of this paper's findings.

3. Lacking literature review (criteria 9, 12). The paper mentions that no "similar approaches" to calving prediction were found, which is fair. But there have been a number of works applying ML to ice sheet dynamics and calving events, and a greater effort could be made to set the context in which this work falls, at the intersection of machine learning and ice sheets. A quick Google Scholar search reveals the following works which in my opinion are quite relevant to the present work and could (should?) be cited:
* Moncada, Francesco. Modeling of ice calving and basal melting in Antarctic region using machine learning. MS thesis. 2024.
* Loebel, Erik, et al. "Calving front monitoring at sub-seasonal resolution: a deep learning application to Greenland glaciers." The Cryosphere Discussions 2023 (2023): 1-21.
* Loebel, Erik, et al. "Calving front positions for 42 key glaciers of the Antarctic Peninsula Ice Sheet: a sub-seasonal record from 2013 to 2023 based on deep-learning application to Landsat multi-spectral imagery." Earth System Science Data 17.1 (2025): 65-78.
I recognise a lot of the work in this space has been related to calving front detection, rather than calving event prediction, but it is worth noting this point in the literature review and only serves to strengthen the contributions of the present work. Additionally: why not use an existing calving front dataset to then train a calving prediction model?

4. Widely divergent range of variable resolutions. This critique concerns Table 1 and the regridding of all variables to 200m resolution. It is not clear how large the tiles are that are considered by the model (see 1.c.) but in any case it is worth noting my concern that the surface mass balance, firn thickness, firn air content, and wind speed and direction variables are nearly 150x more coarse than the ice velocity variable and the ultimate model resolution of 200m. This effectively means that one pixel of e.g. wind speed is interpolated to 155 pixels of length 200m. Depending on the window size considered by the U-Net, does this not effectively make wind speed (and other coarse variables) a constant value input to the model? Or, at least a field with very little variability across the spatial scales considered? It is interesting (concerning?) to note that basal melt rate (1000m resolution) and ice velocity (200m resolution) are the two highest resolution products and are also found to be the most significant predictors; I do not doubt the physical basis of this connection, but does it not draw into doubt how much this result is simply due to these variables being the only variables with significant variability at small spatial scales? To the authors' credit, the issue of coarse/fine variables is mentioned on a few occasions, but I worry this is a very significant modelling flaw which could significantly affect the paper's findings.

5. Discussion of things that have not been done. It's fine (admirable!) to note what work is next in the Future Work section. However, it is downright misleading to identify Pine Island glacier in Figure 1 as an area of study when in fact no results are presented for that region in this current work. It is also downright misleading to mention the use of GradCAM and XAI techniques when these are not used yet in the present work. Keep the discussion of things that have not been done within the Future Work section, and revise the introduction to more accurately reflect the contributions of the present work.\

6. Figure 6 lack of x-axis label. This is a small detail but significantly affects the conclusions one may draw from Figure 6. What is the x-axis representing in Figure 6? Training step/epoch? Or initialisation time in the valiation set? Without an x-axis label, this plot may as well be a bar chart comparing the different models considered. If the x-axis does represent training step/epoch, there should be a good reason to show the line plot over training rather than just the final results; the reason right now is not clear to me, other than to illustrate relatively normal loss convergence and training dynamics.

7. Probability of calving maps (Figures 4, 5, and appendix) do not show the ground truth calving line of Larsen-C, which would be very useful to include to show the reader where the calving actually happened vs where the calving was predicted.

8. Small things.
8.a. Line 017: Weddel Sea should be Weddell Sea
8.b. Line 473: "This indicates that our central hypothesis of approximating dynamical systems remains unfalsified" it would be good to state this as a hypothesis earlier in the work if one wants to claim that it was then not falsified in the conclusion; this hypothesis is not mentioned explicitly.
8.c. Inconsistent use of past/present/future tense throughout the paper. Sometimes "was" is used, sometimes "is" is used, sometimes "will" is used.
8.d. The mention of Lancaster University Centre of Excellence in Environmental Data Science and Centre for Polar Observation and Monitoring is probably a slight violation of double-blind peer review, but not particularly concerning. Just a lesson for the future to remove institutional affiliations like this.

**Justification:**

While I am excited about the important application considered in this paper (predicting ice sheet calving events) and acknowledge that the findings of particular variables (ice velocity and basal melt rate) as important in this predictive task to be physically consistent with known processes (as verified by glaciologist experts), I am very concerned with the experimental rigor of this paper and the clarity of its methodological presentation. If I were given this paper and asked to reproduce it, I could not come close. There is not sufficient detail regarding input variable selection, preprocessing techniques, input image tiling, model architecture choices, training hyperparameter configurations, train/test splitting of data, evaluation techniques, and no code or data is provided by the authors. While a lack of reproducibility may not be grounds for rejection alone, the transparency in experimental methodology is so lacking that I do not know what to make of the results and how to judge the validity of the paper's findings. A significant overhaul of the paper, with much added detail on the points raised above, would be necessary for me to recommend this paper's acceptance. The findings are interesting and the paper does make a contribution to new knowledge, but at this stage I do not feel I can confidently accept the contributions due to the lack of details in the paper's methodological presentation.

---

> ### Author Rebuttal · Authors · 2025-10-20
>
> Thank you for this thorough review, and your further notes on how to improve.
> The GRF data cube will be published at a later stage, and it is our intention that the results should be reproducible at that point.
> As such we will add a section that further describes the architecture in a reproducible manner.
>
> ## Weaknesses
> ### [1] - data preprocessing
> - (a) - The GRF data cube that we sample from will be subject of another paper, as well as a future public release, where the details of the process will be covered more in depth.
> - (b) - We have not applied a normalisation scheme to the data, as the values are not particularly large (e.g. for ice-velocity the values are within -0.5 and 2). We will update the paper to reflect this more clearly along with the next point.
> - (c) - We tested different sizes for the samples, ranging from 128x128 to 256x256. Final sizes were 367x367 over a grid with stride 64x64, in order to rotate and crop the samples to a target of 256x256 (which was the input to the network). We will add a section describing this better.
> - (d) - The train/test split is both spatial and temporal. Training was performed on approximately 60-70% of the total ice sheet, with the remainder being used for validation, and with the test-data produced on an unseen temporal slice of the data cube. In Figure 3 the training and validation split can be seen, where we elected for a simpler hold-out scheme that should align with current practices in the geospatial domain.
> - (e) - We can provide partial or pseudo-code that describes the processes, though a full release of the code may be prohibited at this stage.
>
> ### [2] - theoretical framework
>   Thank you for the feedback. We will replace most of the section on dynamical systems with further detail on the methodological implementation / experimental setup, as well as a fuller Related Work-section.
> ### [3] - literature review
>   We were considering the inclusion of calving front positions over the Antarctic, but ended up electing against it due to the poor resolution of these products. Inspection of the calving fronts show a lot of artifacts that also hinders any form of algorithmic generation of labels. This was our reason for using the calving dataset by Qi et al. The paper will be revised to highlight this data selection better, and justify our non-usage of calving fronts.
>   The thesis from Moncada was released after this project was started, and seems to have a null-result, however in full transparency we will include this as a mention of earlier attempts at similar work.
>   Calving fronts in Greenland are unfortunately also in the same category. The models used are similar in form, but different in target. We can elucidate further on this in the reasoning for the task / data selection.
> ### [4] - resolutions
>   The tiles considered will be adressed (ref 1.c). It also seems clear that the concerns you have regarding the smb, firn-thickness / air-content, and wind-speed are empirically justified in our tests.
>   As you also suggest, the resampling means the coarsest forms of data becomes close to a constant value across a sample. We did not expect these to have a significant contribution, but included them as they were (a) available, and (b) not immediately dismissable as a supporting factor to other variables in the data cube.
> ### [5] - discussion and future work
>   Noted. We will remove the indication of Pine Island as a part of the area of study from Figure 1.
>   While PI is part of the project, it was not part of this study.
> ### [6] - Figure 6
>   The x-axis in Figure 6 is steps. We will rectify this one so that it is properly labeled.
>   The intention of including Figure 6 at all was to justify how the Validation F1 score of wind-speed most likely is a case of overfitting, and not an actual correlation
> ### [7] - Calving overlay
>   We will addd the outline of the actual calving map to Figures 4, 5, and appendix.
> ### [8] - Corrections
> - (a) Thank you, we will correct the language,
> - (b) We will clarify and state our hypotheses early.
> - (c) We will revise and correct the tenses across the paper for final revision.
> - (d) My apologies, I should have done better. We will pay extra attention in future double-blind submissions.
> ## Final remarks
> Thank you again for the feedback, we will revise the paper, focusing specifically on expanding the methodological presentation.
> The input variables are all taken from the available ECVs (essential climate variables) of GCOS, though they are resampled with the GRF, which makes reproduction a bit harder before this data is published at some point next year.
> The data is stacked along with the georeferenced labels, and tiles are randomly picked from a sampling grid (grid size is 64x64), with a padded sampling window to allow data-augmentation. Input and labels are randomly rotated within the range and flipped horizontally or vertically with a $50$ % probability each. The data is then clipped to 256x256, before being trained on. We use an AdamW with $\beta_1 = \beta_2 = 0.9$, learning rate $\lambda = 1e-3$, and a tunable weight-decay starting at $0.001$. The labels are from Qi et al., but did require some manual correction which we will clarify in the revision. We will include a complete model-parameter / hyper-parameter table to the revision to further aid in the reproducability.

---

### Meta-Review · Area_Chair_w7WN · 2025-11-03

**Recommendation:** Accept (Poster)
**Confidence:** 4

**Metareview:**

While the initial paper was raising many issues, all reviewers found the rebuttal very convincing and congratule the authors for having taken this step very seriously. Assuming the authors will revise their manuscript accordingly, the reviewers find the application interesting enough to deserve acceptance (as poster) in NLDL, and are confident that the initial lack of rigour, clarity, and correctness will be corrected in the final version.

---

### Decision · Program_Chairs · 2025-11-05

**Decision:**

Accept (Poster)

**Comment:**

We recommend a poster presentation given the AC and reviewers recommendations.